# Foetal Haemoglobin as a Marker of Bone Marrow Suppression Secondary to Anti-Kell Alloimmunisation

**DOI:** 10.3390/ijns9020024

**Published:** 2023-04-23

**Authors:** Rodrigo Alfredo Morales Painamil, José Manuel González de Aledo-Castillo, Marta Teresa-Palacio, Ana Argudo-Ramírez, Rosa M. López-Galera, Abraham J. Paredes-Fuentes, Victoria Aldecoa-Bilbao, Miguel Alsina-Casanova

**Affiliations:** 1Department of Paediatrics, Hospital Sant Joan de Déu, Esplugues de Llobregat, 08950 Barcelona, Spain; 2Inborn Errors of Metabolism-IBC Section, Biochemistry and Molecular Genetics Department, Hospital Clínic de Barcelona, 08036 Barcelona, Spain; 3Department of Neonatology, BCNatal, Seu Maternitat, Hospital Clínic de Barcelona, 08028 Barcelona, Spain

**Keywords:** transfusion, erythropoietin, hemolytic disease of the newborn, Kell determination, blood group incompatibility

## Abstract

Anti-Kell alloimmunisation is a potentially severe minor blood group type incompatibility, not only as a cause of haemolytic disease of the foetus and newborn, but also due to the destruction of red blood cells (RBC) and mature form in the bone marrow with the subsequent hyporegenerative anaemia. In severe cases and when the foetus shows signs of anaemia, an intrauterine transfusion (IUT) may be necessary. When repeated, this treatment can suppress erythropoiesis and worsen the anaemia. We report the case of a newborn who required four IUTs plus an additional RBC transfusion at one month of life due to late onset anaemia. The identification of an adult haemoglobin profile with a complete absence of foetal haemoglobin in the patient’s newborn screening samples at 2 and 10 days of life warned us of a possible late anaemia. The newborn was successfully treated with transfusion, oral supplements and subcutaneous erythropoietin. A blood sample taken at 4 months of life showed the expected haemoglobin profile for that age with a foetal haemoglobin of 17.7%. This case illustrates the importance of a close follow-up of these patients, as well as the usefulness of the haemoglobin profile screening as a tool for anaemia assessment.

## 1. Introduction

The most common cause of severe haemolytic disease of the foetus and newborn (HDFN) is the presence of maternal alloantibodies to D-Rhesus (RhD) antigens, with K-antibodies second in importance [1]. Despite not being the most frequent cause of severe HDFN, anti-K alloimmunisation is not less important. The most frequent mechanism of immunisation is through previous blood transfusions. Most of these cases are detected by the antenatal indirect antiglobulin test; nevertheless, in most countries, antenatal routine screening and pre-transfusion screening only tests for RhD and ABO antigen, leaving minor groups such as K antibodies unnoticed [2]. Recommendations for antenatal follow-up (antibodies titer threshold, as well as the timing of a Doppler ultrasound follow-up) are tighter in anti-Kell alloimmunisation compared to other alloimmunisations: a different underlying pathophysiology results in an earlier and more severe development of anaemia [3]. When alloimmunisation is detected, maternal antibody titers should be monitored. A lower titer threshold is recommended for anti-K antibodies (anti-K > 1:4 versus non-anti-K > 1:16) due to the earlier development of anaemia [1]. If the titer is above that threshold or there was a previous pregnancy affected by HDFN, signs of foetal anaemia should be excluded with Doppler ultrasound and MCA-PSV measurement [4,5]. When it is over 1.5 MoM, cordocentesis is performed to check the level of anaemia and an intrauterine transfusion (IUT) may be required [4,5,6]. A Doppler ultrasound follow-up of these high-risk pregnancies is recommended from week 18 of gestation versus from 22–24 weeks in non-alloimmunised pregnancies [1]. The main available antenatal therapy in severe cases is IUT, which when repeated, can worsen the anaemic state of the foetus: it can also have an affect postnatally due to the bone marrow suppressive effect of transfusion [7]. We present the case of a newborn who required four IUTs and “top-up” RBC transfusion at one month of life due to late hyporegenerative anaemia. In the newborn screening at 48 h of life, he showed an adult haemoglobin profile, with complete absence of foetal haemoglobin (HbF). A repeated sample was withdrawn at 10 days of life with similar results. He was treated with iron and folic acid supplementation, as well as subcutaneous erythropoietin, with a gradual and progressive recovery of his anaemic state. The haemoglobin profile was normal at 4 months of life. We not only want to highlight the importance of a close follow-up of patients affected by this particular type of alloimmunisation, but also the novelty and usefulness of the haemoglobin profile screening as an extra tool within the assessment of this disease.

## 2. Case Presentation

A case of a term male newborn with late hyporegenerative anaemia (HA) is presented. He was born from the second pregnancy of a 23 year old Caucasian woman. Antenatal screening in the first trimester showed maternal blood group O, Rhesus (Rh) negative, with a positive indirect antiglobulin test (IAT) and the presence of anti-Kell (K) antibodies. Due to the risk of foetal haemolysis and anaemia, serial Doppler ultrasounds were performed to monitor middle cerebral artery peak systolic velocity (MCA-PSV). When this measurement was greater than 1.5 MoM (multiples of the median), a cordocentesis was indicated to check the severity of the anaemia. The identified foetal blood group was O, Rh negative. Four IUTs were required between 24 and 33 weeks of gestational age. There were no signs of hydrops. At 37 weeks labour was induced, but an emergency c-section was performed due to foetal bradycardia. A male newborn was born in good condition, not requiring resuscitation. Apgar score was 9/10, umbilical artery pH 7.24, with a negative direct antiglobulin test (DAT). The patient did not present signs of haemolysis. He was discharged home at 48 h of life with a haematocrit of 40% and a bilirubin of 6.2 mg/dL.

Since 2015, the newborn screening program in Catalonia has included the analysis of haemoglobin fractions by capillary electrophoresis (Capillarys 2, SEBIA, Lisses, France) to detect sickle cell disease [8]. This analysis in the patient showed an adult haemoglobin profile with 98.1% of haemoglobin A (Hb A), 1.9% of haemoglobin A2 (Hb A2) and complete absence of foetal haemoglobin (HbF) (shown in Figure 1). A repeated sample at 10 days of life demonstrated similar results (shown in Figure 2).

Follow-up at 29 days of life showed adequate weight gain and normal physical examination but pallor. Haematocrit at this time was 14%, haemoglobin 4.9 g/dL and reticulocytes 2.6%. He received a single packed red blood cell transfusion and his post-procedure haematocrit was 33% and haemoglobin 11.4 g/dL. He was started on iron and folic acid supplements and followed up with as an outpatient every 10 days (shown in Table 1). 

Due to a further drop in haematocrit with a still low level of reticulocytes, subcutaneous recombinant human erythropoietin (rhEPO) was started at 51 days of life and administered three times weekly. The patient finally reached stability of blood parameters around 70 days of life. A blood sample taken at 4 months of life showed the expected haemoglobin profile for that age (shown in Figure 3).

## 3. Discussion

In spite of not being the most frequent cause of severe HDFN, anti-K alloimmunisation can be troublesome both pre and postnatal. The main mechanism of anaemia in alloimmunisation is the destruction of peripheral RBCs. In anti-K immunisation, however, the HA plays a more relevant role than haemolysis. K antigens are expressed by RBC precursors and mature cells from early foetal development. Anti-K antibodies cause the destruction of these cells and inhibit the growth of erythroid progenitor cells from then, causing an earlier HA than other types of HDFN [2,3]. The patient in this case presented anaemia with a low reticulocyte count and normal bilirubin levels supporting the pathophysiology of this type of alloimmunisation.

IUT is the only antenatal available therapeutic option to improve the perinatal outcome [6]. However, it can lead to a suppression of foetal erythropoiesis, leading to a HA that worsens the already established one [7]. This happens especially if more than three IUTs are required [9]. Late HA mostly affects newborns with a history of IUT, early anaemia after birth and severe haemolytic jaundice. Patients with these characteristics should be assessed during the immediate newborn period and adequately followed-up with.

Around 50% of newborns who received an IUT will require a “top-up” RBC transfusion, with an increased number of postnatal transfusions needed as the more IUTs have been received. Outpatient treatment with rhEPO or darbepoetin during 4 to 6 weeks can reduce transfusional requirements [10,11]. These treatments do not produce an immediate stabilisation of haemoglobin, as it takes about a week. All patients should also receive iron and folic acid supplementation. This supplementation should be adjusted by serum iron, ferritin and transferrin saturation tests, as was done in the current case. HA and also haemolytic anaemias can sometimes be associated with iron overload, rather than iron deficiency. Postnatal follow-up of patients at risk of late HA should be scheduled until the blood parameters remain stable for at least 3 weeks. Perinatal survival rates with appropriate clinical management approach 97% [12].

A close follow-up plays a big role in prompt identification of late onset HA. We believe that through this case, an early screening of the haemoglobin profile can be useful to predict the severity of the HA that the patient may develop. A blood test of the patients who have undergone IUT can show an adult haemoglobin profile; however, in this case a total absence of HbF was found, which is very unusual [13,14]. Because of the predominance of the donor’s blood, these patients can have a negative DAT, as it was in our patient. This should have been assumed as a high risk of a worsening anaemia, and made us schedule an earlier follow-up with blood sampling. The total absence of HbF has to be interpreted as the lack of all the newborn’s own RBCs and the only presence of adult donor cells. The progressive destruction of the donor’s cells added to the presence of maternal antibodies and of HA, all together, may contribute to the development of clinically significant anaemia which would require a closer follow-up.

## 4. Conclusions

This case report shows the special characteristics of anti-K alloimmunisation. The specific pathogenesis of anaemia and the treatment with IUT which worsens the already established anaemia, reflect the importance of adequate follow-up with these patients both antenatal and postnatally. Along with this, we think that the analysis of the haemoglobin profile with newborn screening can be a very useful tool to predict the severity of the HA that the patient may develop.

## Figures and Tables

**Figure 1 IJNS-09-00024-f001:**
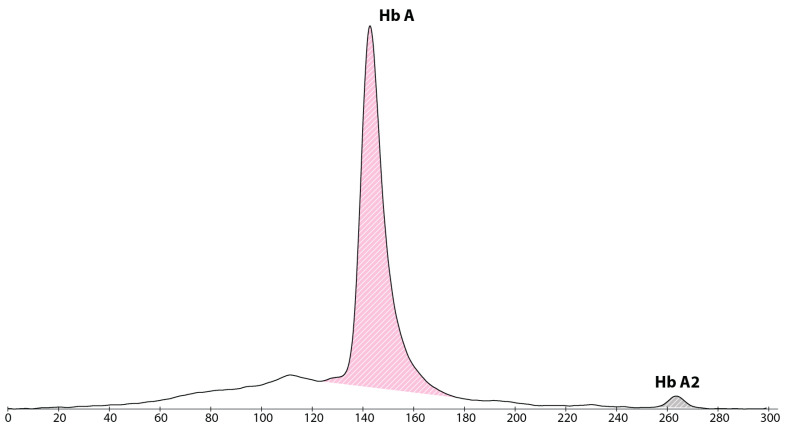
Haemoglobin electropherogram at 48 h of life, showing a major peak of haemoglobin A, absence of haemoglobin F and a small peak of haemoglobin A2.

**Figure 2 IJNS-09-00024-f002:**
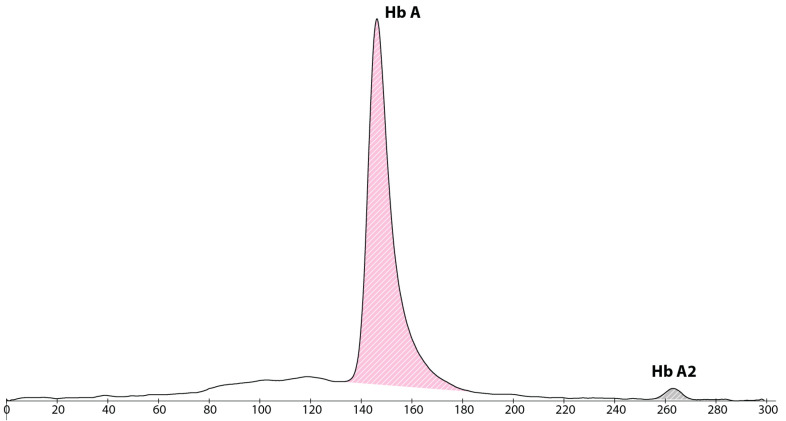
Haemoglobin electropherogram at 10 days of life, showing a major peak of haemoglobin A, absence of haemoglobin F and a small peak of haemoglobin A2.

**Figure 3 IJNS-09-00024-f003:**
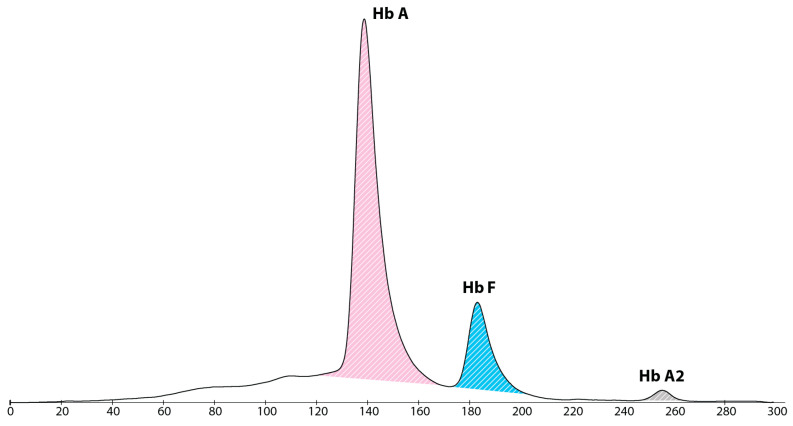
Haemoglobin electropherogram at 4 months of life, showing a major peak of haemoglobin A, presence of haemoglobin F and a small peak of haemoglobin A2.

**Table 1 IJNS-09-00024-t001:** Follow-up summary of our patient. PT = phototherapy; EPO = recombinant human erythropoietin; RBC = red blood cell; HbF = foetal haemoglobin.

	Age (days)	0–2	29	30	41	51	64	80	120
Parameter									
Weight (grams)		2900–2720	3632	3632	4100	4550	5050	5490	6700
Haematocrit (%)		40	14	33	25	22	21	28.4	34.7
Haemoglobin (g/dL)		12.1	4.9	11.4	8.8	7.7	7.3	9.3	12.2
HbF (%)		0							17.7
Reticulocytes (%)			2.6		1.5	3.2	7.3	8.6	1.11
Total Bilirubin (mg/dL)		2.7–6.2	2.3						
Treatment		PT	20 mL/kg RBC transfusion	None	Oral iron and folic acid	+EPO thrice weekly	Same	STOP EPO Adjustment of oral iron	Same

## Data Availability

The data presented in this study are available in entirety within this article.

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
