# Peer review of "Foetal Haemoglobin as a Marker of Bone Marrow Suppression Secondary to Anti-Kell Alloimmunisation"

_2409-515X, 2023, doi:10.3390/ijns9020024_

Round 1
Reviewer 1 Report
Dear Authors,
Some minor corrections that could improve the quality of this document:
1. The introduction is very short. In my opinion, it should be expanded on the part of the main causes leading to hemolytic complications of the fetus and the newborn. Thus the paper will be more understandable to a wider audience.
2. It would be helpful if data on hematocrit, hemoglobin, and reticulocyte levels in the study periods (i.e., before and after the administered treatment) should be given in the table.
3. The first two paragraphs (from line 97 to line 110) of the "Discussion" section are better suited to the "Introduction" section and should be moved there.
After these corrections, I will propose publishing it in the International Journal of Neonatal Screening.
Author Response
Dear reviewer,
Thank you very much for reviewing the paper and for your suggestions to amend it.
- The introduction is very short. In my opinion, it should be expanded on the part of the main causes leading to hemolytic complications of the fetus and the newborn. Thus the paper will be more understandable to a wider audience.
With respect to the correction number one, we edited the introduction paragraph to give a bigger idea of the issue addressed afterwards.
- It would be helpful if data on hematocrit, hemoglobin, and reticulocyte levels in the study periods (i.e., before and after the administered treatment) should be given in the table.
As for the second correction, we added a table with the weight and blood sampling follow-up of our patient, alongside with treatment changes.
- The first two paragraphs (from line 97 to line 110) of the "Discussion" section are better suited to the "Introduction" section and should be moved there.
As for the third correction, we took the cited lines and put them within the introduction section so it would broaden up the explanation intro for the topic addressed.
Reviewer 2 Report
The authors describe a full-term male newborn with late hypo regenerative anaemia (HA) that was born from the second pregnancy of a 23 years old Caucasian woman. Antenatal screening in the first trimester showed maternal blood group O, Rhesus (Rh) negative, with a positive indirect antiglobulin test (IAT) and the presence of anti-Kell (K) antibodies.
Comments:
This is an interesting clinical observation of neonatal rare anaemia where the importance of the newborn screening program in Catalonia that includes the analysis of haemoglobin fractions by capillary electrophoresis is reinforced. The paper is technically sound and well-written, but it should be clearly indicated that the administration of iron supplements was supported by the demonstration of iron deficiency as provided by the serum iron, ferritin and TSI tests. Hypo regenerative (HR) anaemias and also hemolytic anaemias, maybe sometimes associated with iron overload, instead of iron deficiency,
Author Response
Dear reviewer,
Thank you for the time taken into reading our article, as well as for the comments and suggestions. The authors describe a full-term male newborn with late hypo regenerative anaemia (HA) that was born from the second pregnancy of a 23 years old Caucasian woman. Antenatal screening in the first trimester showed maternal blood group O, Rhesus (Rh) negative, with a positive indirect antiglobulin test (IAT) and the presence of anti-Kell (K) antibodies.Comments:
This is an interesting clinical observation of neonatal rare anaemia where the importance of the newborn screening program in Catalonia that includes the analysis of haemoglobin fractions by capillary electrophoresis is reinforced. The paper is technically sound and well-written, but it should be clearly indicated that the administration of iron supplements was supported by the demonstration of iron deficiency as provided by the serum iron, ferritin and TSI tests. Hypo regenerative (HR) anaemias and also hemolytic anaemias, maybe sometimes associated with iron overload, instead of iron deficiency.
Patients that have received a "top-up" transfusion after birth, a proper recovery from the anaemic state as well as the erythrocite production will be achieved with adecuate iron supplementation afterwards. This is specially important in pacients that receive erythropoiesis stimulating agents, because its effect will be affected by a correct iron store and supplementation. Nevertheless, we agree that those mentioned types of anaemia are sometimes associated with iron overload instead of deficiency, and thus we have added your suggestions to the text.